# Polish Adaptation of the *Alarm Fatigue Assessment Questionnaire* as an Element of Improving Patient Safety

**DOI:** 10.3390/ijerph20031734

**Published:** 2023-01-18

**Authors:** Łukasz Rypicz, Anna Rozensztrauch, Olga Fedorowicz, Aleksander Włodarczyk, Katarzyna Zatońska, Raúl Juárez-Vela, Izabela Witczak

**Affiliations:** 1Department of Population Health, Division of Public Health, Faculty of Health Sciences, Wroclaw Medical University, 51-618 Wroclaw, Poland; 2Department of Nursing and Midwifery, Faculty of Health Sciences, Wroclaw Medical University, 51-618 Wroclaw, Poland; 3Department of Clinical Pharmacology, Faculty of Pharmacy, Wroclaw Medical University, 51-618 Wroclaw, Poland; 4Faculty of Medical Sciences named after Professor Zbigniew Religa, Academy of Silesia, 40-007 Katowice, Poland; 5Research Group GRUPAC, Faculty of Health Sciences, University of La Rioja, 26004 Logroño, Spain

**Keywords:** alarm fatigue, ergonomics, patient safety

## Abstract

Medical personnel, working in medical intensive care units, are exposed to fatigue associated with alarms emitted by numerous medical devices used for diagnosing, treating, and monitoring patients. Alarm fatigue is a safety and quality problem in patient care and actions should be taken to reduce this by, among other measures, building an effective safety culture. In the present study, an adaptation of a questionnaire to assess alarm fatigue was carried out. The study obtained good reliability of the questionnaire at Cronbach’s alpha level of 0.88. The Polish research team has successfully adapted the Alarm Fatigue Assessment Questionnaire so that it can be used in healthcare settings as a tool to improve patient safety.

## 1. Introduction

Medical personnel, particularly working in medical intensive care units, is exposed to fatigue with alarms emitted by numerous medical devices used for diagnosing, treating, and monitoring patients [1,2]. Some of these alarms go off in the absence of an intended important event, some when the alarm system is working properly but indicate an event that is not clinically relevant and/or does not require additional intervention [3]. Alarm fatigue is a type of sensory overload that can lead to indifference to or overlooking the emitted signals [4]. Bedside monitors with alarms usually monitor patient’s physiological parameters such as heart rate and arrhythmia, respiratory rate, indirect measurement of blood oxygen saturation by peripheral pulse oximetry, or invasive monitoring of arterial, intracranial, and central venous pressure parameters [5]. Electronic medical devices are an indispensable part of patient care, and the number of alarms during a doctor’s or nurse’s shift can reach up to 1000 alarms per shift [6]. It is reported that between 72% and 99% of clinical alarms may be false [4,7,8].

The World Health Organization (WHO) Guidelines for Community Noise indicates that noise levels in hospitals should not exceed 35 dBA [9,10]. However, studies show that noise levels in hospital environments range from 47 to 77 dBA. These noise levels can disrupt the workflow of medical staff, contribute to alarm fatigue which may results in inappropriate practices in responding to alarms, and ultimately lead to errors in patient care. This, in turn, adversely affects staff cooperation, increases aggression, and impairs the ability to process social signals. Noise is also a factor that contributes to an increased sense of fear and stress in a hospital ward [11,12]. 

In 2014, alarm fatigue was recognized as a serious problem in ensuring patient safety and was entered into the National Patient Safety Goals on Alarm Management as a safety issue that needs improvement [4,6]. It appears that overexposure of medical staff to the noise generated by alarms from medical devices can reduce patient safety [13,14]. Indifference to alarms or fatigue from false or unnecessary alarms can lead to serious adverse events and even patient death [15]. From 2005 to 2008, the US Food and Drug Administration (FDA) received 566 reports of patient deaths that were related to monitoring device alarms and in 2010, it received more than 2500 adverse event reports related to medical devices, of which nearly one-third concerned problems with the alarm system [8]. Alarm fatigue is a safety and quality problem in patient care and actions should be taken to reduce this by, among other measures, building an effective safety culture. This involves, among other things, establishing safe alarm management and response processes [8]. 

A review of the literature on alarm fatigue and patient and medical staff safety indicates that it is still a major problem that requires further in-depth research. Looking at the scale of the problem, it seems reasonable to adapt a questionnaire to assess alarm fatigue among medical staff. The diagnosis of the problem can provide a basis for taking preventive action and improving the safety of patients and medical staff. In view of the above, the authors of this article attempted a cross-cultural adaptation of a questionnaire to assess alarm fatigue. 

This study aimed to produce a Polish adaptation of the Alarm Fatigue Assessment Questionnaire to assess alarm fatigue among healthcare workers in anesthesiology and intensive care units.

## 2. Materials and Methods

This cross—sectional psychometric validation was conducted from June to September 2021. A total of 103 respondents participated in the study. Sample collection was performed in leading hospitals with Pediatric and Adults Intensive Care Units. It should be clarified that in the Polish health care system there are departments which are called “Department of anesthesiology and intensive care”. Anesthesiology was not excluded from the study.

### 2.1. Translation, Adaptation, and Modeling

The translation and cultural adaptation procedure was performed in accordance with the international standards described by Beaton et al. and Wild et al. [16,17]. The whole process consisted of six stages: (1) initial translation, (2) synthesis of translations, (3) back translation, (4) expert committee evaluation, (5) testing the draft version, and (6) submission of the final version to the developers or Coordinating Committee for the Evaluation of the Adaptation Process. The questionnaire was translated from English into Polish by two independent sworn translators specializing in medical and health sciences. The translators received guidance from the research team to avoid metaphors and use simple, understandable phrases. Both translations were then assessed by the research team, who are fluent in English and familiar with the medical terminology and background of this study. During this session, all translation differences were discussed, and the first Polish-language version of the questionnaire was obtained. This first Polish-language version of the questionnaire was re-translated into English—the translation was done by an independent native speaker who was not familiar with the source version. A panel of experts with multidisciplinary qualifications and experience in healthcare management, nursing, medicine, public health, and work ergonomics determined the final version of the questionnaire. All members of the panel of expert work professionally at the university and in hospitals, which translates into extensive practical and theoretical experience. In addition, members of the panel of expert have experience in conducting adaptations of research tools culminating in publications. In the next step, the questionnaire was tested on a sample of 15 nurses, who were asked to confirm that all questions were understandable, unambiguous, and not questionable. This task was to confirm that the questionnaire was prepared correctly and could be used for the survey. During the questionnaire test, none of the questions were challenged. Therefore, all questions were included in the study.

### 2.2. Participants and Settings

Following the purposeful selection, invitations to participate were received by medical employees of wards in which there are devices emitting alarms. A flyer with information about the survey and an envelope with a survey questionnaire were distributed to hospital departments. Inclusion criterion involved physicians, nurses, and paramedics who were active practitioners, providing care to patients with medical equipment working in units imitating alarms. Respondents were informed that participation in the study was voluntary and anonymous, and they were informed about the purpose of the study. Only correctly completed questionnaires were included in the psychometric analysis. All data were collected anonymously. Study participants, after receiving full instructions, completed a one-time questionnaire that was a Polish translation of the full version of the original Alarm Fatigue Assessment Questionnaire. The completed questionnaires were placed in an envelope and left at the secretariat from where members of the research team collected them and a database was prepared from them, which was used in the statistical analysis.

### 2.3. Alarm Fatigue Assessment Questionnaire

The questionnaire for the assessment of alarm fatigue, after adaptation into Polish, was published in 2017 by Ashrafi et al. [18]. The tool consists of 23 items (Table 1) formulated as statements for which respondents specified frequency using a 5-point Likert scale: always (5), usually (4), sometimes (3), rarely (2), and never (1). Items identified the perceptions of medical staff about alarm fatigue.

### 2.4. Ethical Considerations

The study was carried out in accordance with the tenets of the Declaration of Helsinki and the guidelines of Good Clinical Practice (World Medical Association, 2013).

Written information about the study was provided as an introduction to the survey, with an emphasis on the voluntary and anonymous nature of participation and its guaranteed confidentiality. By answering the questionnaire, participants gave their consent to participate in the study. The research project was approved by the independent Bioethics Committee at the Wroclaw Medical University (No KB–384/2021).

### 2.5. Statistical Analysis

The reliability of the scale used was checked by calculating Cronbach’s alpha coefficient for individual items. Cronbach’s alpha (α) values should optimally range between the recommended values of 0.60–0.90. The following thresholds for internal consistency were used: 0.9 ≤ α—excellent; 0.8 ≤ α < 0.9—good; 0.7 ≤ α < 0.8—acceptable; 0.6 ≤ α < 0.7—questionable; 0.5 ≤ α < 0.6—poor; and α < 0.5—unacceptable. The statistical analysis was carried out by a professional statistician. The analysis was performed in the R program, version 3.6.1 (R Foundation for Statistical Computing: Vienna, Austria) (R Core Team, 2019) [19].

An item-by-item analysis was carried out based on the percentage floor effect and ceiling effect. In addition, a confirmatory factor analysis was performed. As the items of the questionnaire used are expressed on an ordinal and not a continuous scale, the Diagonally Weighted Least Squares method was used. For the univariate (i.e., assuming no subscales) structure, fit indices were calculated.

## 3. Results

The socio-demographic data of the sample are presented in Table 2. On the basis of the results obtained, it should be noted that the sample was strongly feminized (female 94%, male 6%). The mean age of respondents was 40.4 years, and the mean length of service was 17.1 years. The vast majority of respondents were nurses (84%), followed by the professional groups of doctors (11%) and paramedics (5%). Of those included in the survey, more than half (56%) have completed a Masters degree (this also includes medical studies leading to a medical degree), while 1/3 of the respondents have completed their education at a Bachelor’s level (30%). The smallest group were those who graduated from high school (14%—they were nurses who graduated from medical high schools—currently, there are no medical high schools in the system of education). The survey was conducted in intensive care units, with just over half of the respondents working in hospital units for children (55%) and the remainder in hospital units for adults (45%). About two thirds worked in more than one place (65%).

### 3.1. Analysis of the Individual Questionnaire Items

Table 3 shows the results of the analysis of the individual questionnaire items. Floor effect is the percentage of respondents who chose the lowest scoring response to a question. Ceiling effect is the percentage of those who chose the highest scoring answer. A high floor or ceiling effect in a question indicates that it may not be well matched to the population being analysed.

In our study, a high floor effect was obtained in question 1, which indicates that a rejection of this question should be considered for the adapted questionnaire. Perhaps it is too obvious for the population studied. A high floor or ceiling effect rate (above 70%) was also obtained for items: 1, 17, and 22. However, due to the relevance of the questions in the context of adverse events (as reported in the literature) that occur with alarm fatigue—it was decided to leave them out.

### 3.2. Confirmatory Factor Analysis

In the validation of a tool, it is always worthwhile carrying out a factor analysis (factor analysis), which will give an answer to the question of whether the tool should be divided into subscales. A distinction is made between exploratory (EFA) and confirmatory factor analysis (CFA). When validating an existing tool (e.g., a language adaptation), there is no need to perform an EFA. As our questionnaire’s items are expressed on an ordinal rather than a continuous scale, the Diagonally Weighted Least Squares method was used.

For the univariate (i.e., assuming no subscales) structure, not entirely satisfactory values of fit indices were obtained—the exact values are described as Model I in the table below (Table 4).

The Chi-square test for well-fitted models should come out non-significant (*p* > 0.05). In the study, we obtained a result where *p* < 0.05, but this should be approached with caution, as a well-known and much-discussed drawback of this test is that it almost always gives a low *p* on large samples (if we have a large test sample, we are unlikely to get satisfactory results in this test [20]). This disadvantage is not present in the other measures presented in the table: the RMSEA (Root Mean Square Error of Approximation, CFI (Comparative Fit Index), TLI (Tucker-Lewis Index), and SRMR (Standardised Root Mean Residual). A number of cut-off point proposals for these measures can be found in the literature. Hu and Bentler [21], in their paper, indicated that the model is a good fit when we have RMSEA < 0.06, CFI and TLI > 0.95, and SRMR < 0.08. However, they point out that meeting the conditions for all four measures is often too restrictive a requirement and they propose a method they call Two-Index Strategy. It says that a model is a good fit when SRMR < 0.09 and additionally one of the conditions CFI > 0.96, TLI > 0.96, or RMSEA < 0.06 occurs. In the present study, these conditions are not met, as an SRMR > 0.09 was obtained.

In the next step, the loadings of the individual items were therefore checked. Loadings are interpreted as correlations of items with the subscale to which they belong or with the total score if the tool has no subscales.

It turned out (loadings column (Model I) in Table 5 that Items 1, 8, and 10 have very low loadings (they are very weakly correlated with the total score). They were therefore removed from the tool.

The fit measures for the tool with items 1, 8, and 10 omitted were still not satisfactory (SRMR > 0.09, Model II in Table 4). In the next step, the modifications indicated by the so-called modification indexes were applied. In this case, they suggest introducing a correlation into the model between the following item pairs: 2 and 15, and 11 and 12. This makes it possible to obtain the desired parameter values (SRMR < 0.09, CFI > 0.96, TLI > 0.96, RMSEA < 0.06, Model III in the Table 4). Thus, in practice, we have confirmation of the univariate structure of our questionnaire (in the version without items 1, 8, and 10) with the caveat that the answers to the extracted questions are strongly correlated with each other. In other words: they ask very similar things.

The absolute values of the item loads in the final model ranged from 0.203 to 0.769 and were statistically significant (*p* < 0.05). The charges for items 2, 3, 7, 13, 14, and 15 have the opposite sign (which are positive) than the charges of the other items (which are negative). This means that items 2, 3, 7, 13, 14, and 15 should be recoded (i.e., the answers “1” in them have to be changed to “5”, “2” to “4”, etc.).

### 3.3. Reliability Analysis of the Tool

Cronbach’s alpha for this tool, after recoding items 2, 3, 7, 13, 14, and 15, is 0.881. The scale is therefore reliable (Table 6). In other words: its results are reproducible, not due to chance. An alpha above 0.7 is assumed to be a reliable scale. All items have positive discriminatory power (Item-Total correlation). This means that they positively correlate with the other items included in the scale, which is a very desirable effect.

## 4. Discussion

The cross-cultural adaptation of the Polish version of the Alarm Fatigue Assessment Questionnaire was carried out successfully. To the authors’ best knowledge, this is the first adaptation of the tool into Polish, which was also the main argument for the need to adapt this tool. The literature review clearly demonstrates the negative impact of alarm fatigue on ensuring patient safety, which is another important rationale for such an adaptation.

There are isolated studies in the literature on adapting an alarm fatigue assessment questionnaire dedicated to nurses. It is a tool developed by Iranian researchers. Its authors achieved a reliability level of the tool with a Cronbach’s alpha value of 0.91 [22]. Lebanese authors in their adaptation of this questionnaire obtained a Cronbach’s alpha of 0.69 [12], while a Turkish adaptation showed a Cronbach’s alpha of 0.71 [23]. It should be noted that against the background of previous adaptations of a similar tool for the assessment of alarm fatigue, the results obtained by the Polish team are very good (Cronbach’s alpha 0.88). Moreover, when analysing each item separately, none of the items obtained a Cronbach’s alpha value below 0.8, which indicates that the Polish version of the questionnaire was prepared according to the protocol used for adaptations.

Thanks to the Polish version of the questionnaire, it will be possible to use the tool in medical institutions for intervention clinical audits, preventive measures against professional burnout in intensive care units or preventive/corrective measures in case of an adverse event related to medical device alarms.

The wide range of questions in the questionnaire about the impact of alarms emitted by medical devices on medical personnel makes it possible to quickly identify areas which may be potential sources of an adverse event. This tool can become a permanent part of a medical facility’s patient safety culture as part of a programme to improve the quality of medical care and workplace ergonomics. The use of the Alarm Fatigue Assessment Questionnaire in clinical risk management can bring tangible benefits to the employer and medical staff. It is believed that one of the basic steps in alarm management is to conduct a baseline alarm assessment to identify current needs and conditions contributing to alarm fatigue [3]. The Polish version of the questionnaire may be an excellent tool for this purpose, especially since the literature review indicates that implementing specific elements of a safety culture can lead to a reduction in the total number of alarms and in the number of false alarms, and a reduction in alarm noise.

Medical device alarm management plays an extremely important role in clinical risk reduction. The main recipients of signals emitted by equipment in intensive care and neonatal units are nursing staff. It is the nurses’ behavior that is key to alarm management. Some hospitals that introduce elements of alarm management run educational programmes to improve nurses’ skills in recognizing the alarm sounds of medical devices and to standardise actions around these devices [24]. In the literature, we can find information that in intensive care units we can locate from 6 to 40 types of alarms emitted by medical devices. The differences between the alarms emitted by infusion pumps and ventilators depend, among others, on the producer—they may differ in volume and sound. If these devices do not differ significantly in alarm volume in clinical settings, nurses’ perceptions of the inaudibility and frequent misses of infusion pump alarms are related to the perceived criticality of these alarms [25].

Several studies also indicate that the involvement of representatives from other disciplines can greatly facilitate the effective assessment of alarm fatigue risk. The aforementioned studies note that a multidisciplinary team including representatives not only from clinical staff but also from computer scientists or biomedical engineers, provides an effective assessment of alarm fatigue. Interdisciplinary experts can assess the current state of the clinical alarm environment, and help establish baseline alarm values by unit, patient group, and time of day or night [26,27,28]. Hospital managers will not solve a nuisance alarm problem if they do not know what the problem is, its risks, or its consequences.

## 5. Conclusions

The presented study shows that no Polish adaptation of a tool investigating alarm fatigue has been published so far. The Polish research team has successfully adapted the Alarm Fatigue Assessment Questionnaire so that it can be used in healthcare settings as a tool to improve patient safety. Alarm fatigue poses a major challenge for managers of healthcare facilities, and previous work points to in-depth research in this area. 

Building a culture of patient and medical staff safety is an indispensable part of everyday life in any healthcare system. One of the many elements of ensuring patient and medical staff safety is the management of medical device alarms, which is closely linked to clinical risk and ergonomic factors in the medical staff workplace. It appears that developing adaptations of tools/questionnaires that can be applied in creating and improving safety culture in healthcare facilities is very important. One more important element should be kept in mind, namely the education of medical personnel and communicating to them about new possibilities and solutions in the given field. Without this element, the implementation of new tools/questionnaires will not make sense, because the recipients of these solutions will not know why they are used. The introduction of such tools is therefore one of the first steps that clinical leaders need to take to gain data about alarms. These will allow them to develop appropriate strategies that seek to reduce alarm fatigue. It should be noted that the adaptation process was carried out mainly on the nursing staff, but on the other hand, it is the nursing staff that is mainly exposed to fatigue alarms when caring for patients.

### Study Limitations

The study also has some limitations. The sample could have been larger, nevertheless, it is sufficient for the required statistical analysis for tool adaptation. Moreover, the study sample is highly homogenous in terms of gender. This is due to the fact that the professional group of nurses in Poland is highly feminized—the percentage of men among the nursing staff in Poland was only 2.67% (data as of the end of 2021). The study did not use an electronic version of the questionnaire; thus, its potential reach was limited. It should also be noted that since the group was largely made up of nursing staff, the application of the tool to another professional group should be performedwith great caution.

## Figures and Tables

**Table 1 ijerph-20-01734-t001:** Individual items from the Alarm Fatigue Assessment Questionnaire.

Item	Text of the item in English and Polish	Acceptance/Rejection
Item 1	I pay attention to the changes of alarm source immediately after hearing the alarm. Po usłyszeniu alarmu zwracam uwagę na to, czy jego źródło się zmieniło.	Rejected
Item 2	I am sure that the alarms are true. Słysząc alarm mam pewność, że jest on prawdziwy.	Accepted
Item 3	I go to the patient’s bed immediately after I hear an alarm. Kieruję się do łóżka pacjenta natychmiast po usłyszeniu alarmu.	Accepted
Item 4	During my shift, I limit the number of alarms. Ograniczam liczbę alarmów podczas swojej zmiany.	Accepted
Item 5	Alarms hinder my focus on professional duties. Alarmy utrudniają mi skupienie się na swoich obowiązkach służbowych.	Accepted
Item 6	I get nervous when I hear an alarm. Gdy słyszę alarm, staję się nerwowy(a).	Accepted
Item 7	I have a proper professional reaction toward alarms. Słysząc alarm reaguję w profesjonalny sposób.	Accepted
Item 8	I try to distinguish the informing alarms (yellow) and warning alarms (red). Staram się rozróżniać alarmamy informacyjne (żółte) od ostrzegawczych (czerwone).	Rejected
Item 9	I stop as I hear the alarm, maybe it is settled by itself. Po usłyszeniu alarmu zatrzymuje się—może problem sam się rozwiąże.	Accepted
Item 10	I pay more attention to the alarms in night shifts. Na alarmy zwracam większą uwagę podczas nocnych zmian.	Rejected
Item 11	In the morning shift, the crowd hinder my immediate reaction to alarms. Tłok w trakcie porannej zmiany utrudnia mi reagowanie na alarmy.	Accepted
Item 12	At the beginning of each shift, I pay more attention to the alarms. Na alarmy zwracam większą uwagę na początku zmiany.	Accepted
Item 13	I have an immediate reaction to the ventilator alarms. Na alarmy respiratora reaguję natychmiast.	Accepted
Item 14	I have an immediate reaction to the infusion pump alarms. Na alarmy pompy infuzyjnej reaguję natychmiast.	Accepted
Item 15	I have an immediate reaction to cardiac monitoring alarms. Na alarmy kardiomonitora reaguję natychmiast.	Accepted
Item 16	In the course of time, my sensitivity to alarms decreases. Z biegiem czasu staję się coraz mniej wyczulony(a) na alarmy.	Accepted
Item 17	I am indifferent to the alarms. Alarmy są mi obojętne.	Accepted
Item 18	During a CPR in a patient, I become indifferent to the alarms of other patients. Wykonując resuscytację na jednym pacjencie staję się obojętny(a) na alarmy innych.	Accepted
Item 19	By repetition of alarms, I become indifferent to them. Ciągłe powtarzanie się alarmów sprawia, że staję się na nie obojętny(a).	Accepted
Item 20	Multiplicity and concurrence of alarms confuse me in making decisions. Mnogość alarmów i ich jednoczesne występowanie w kilku miejscach sprawiają, że trudno jest mi podjąć decyzję.	Accepted
Item 21	I do not pay attention to the alarm when I do not feel well. Gdy źle się czuję, nie zwracam uwagi na alarmy.	Accepted
Item 22	I inactivate the alarms in the night shifts. Podczas nocnej zmiany wyłączam alarmy.	Accepted
Item 23	I become confused with successive sounds of alarms. Powtarzające się raz po raz alarmy wywołują u mnie zdezorientowanie.	Accepted

**Table 2 ijerph-20-01734-t002:** Socio-demographic characteristics of the study sample (N = 103).

Parameter	Total (N = 103)	
Sex	Female	97 (94.17%)
	Male	6 (5.83%)
Age [years]	Mean (SD)	40.4 (11.05)
Seniority [years]	Mean (SD)	17.14 (11.55)
	Median (quartiles)	17 (5–28)
	Range	1–38
	Missing	0
Number of workplaces	One	36 (34.95%)
	More than one	67 (65.05%)
Profession	Physician	11 (10.68%)
Nurse	87 (84.47%)	
Paramedic	5 (4.85%)	
Education	Secondary	14 (13.59%)
Bachelor’s degree	31 (30.10%)	
Master’s dergree	58 (56.31%)	
Unit	ICU—adults	46 (44.66%)
ICU—children	57 (55.34%)	

ICU—Intensive Care Unit, Abbreviations: M—mean, Me—median, min—minimum, max—maximum, SD—standard deviation.

**Table 3 ijerph-20-01734-t003:** Analysis of the individual questionnaire items.

Item	Floor Effect	Ceiling Effect	Missing
1	78.6%	0.0%	0.0%
2	26.2%	1.0%	0.0%
3	54.4%	0.0%	0.0%
4	11.7%	38.8%	0.0%
5	10.7%	11.7%	0.0%
6	5.8%	15.5%	0.0%
7	49.5%	0.0%	0.0%
8	56.3%	0.0%	0.0%
9	2.9%	27.2%	0.0%
10	28.2%	19.4%	0.0%
11	3.9%	15.5%	0.0%
12	8.7%	22.3%	0.0%
13	68.9%	0.0%	0.0%
14	56.3%	0.0%	0.0%
15	56.3%	0.0%	0.0%
16	5.8%	30.1%	0.0%
17	3.9%	75.7%	0.0%
18	7.8%	33.0%	0.0%
19	3.9%	28.2%	0.0%
20	2.9%	34.0%	0.0%
21	1.9%	48.5%	0.0%
22	1.9%	77.7%	0.0%
23	3.9%	23.3%	0.0%

**Table 4 ijerph-20-01734-t004:** Results of fit indicies.

Model	Chi-Squared Test			RMSEA *	CFI *	TLI *	SRMR *
	χ²	df	*p*				
Model I	281,559	230	0.011	0.047	0.963	0.959	0.101
Model II	162,156	170	0.654	>0.001	>0.999	>0.999	0.093
Model III	142,759	168	0.922	>0.001	>0.999	>0.999	0.088

* RMSEA—Root Mean Square Error of Approximation, CFI—Comparative Fit Index, TLI—Tucker-Lewis Index i SRMR -Standardized Root Mean Residual.

**Table 5 ijerph-20-01734-t005:** Results of load values of individual items.

Item	Loadings (Model I)	Loadings (Model III)	*p* (Model III)
1	−0.009	---	---
2	0.232	0.203	*p* = 0.029
3	0.222	0.224	*p* = 0.021
4	−0.579	−0.579	*p* < 0.001
5	−0.455	−0.454	*p* < 0.001
6	−0.642	−0.643	*p* < 0.001
7	0.216	0.22	*p* = 0.047
8	0.001	---	---
9	−0.586	−0.587	*p* < 0.001
10	−0.053	---	---
11	−0.586	−0.568	*p* < 0.001
12	−0.281	−0.252	*p* = 0.018
13	0.385	0.388	*p* < 0.001
14	0.273	0.274	*p* = 0.002
15	0.489	0.472	*p* < 0.001
16	−0.718	−0.724	*p* < 0.001
17	−0.686	−0.688	*p* < 0.001
18	−0.593	−0.596	*p* < 0.001
19	−0.766	−0.769	*p* < 0.001
20	−0.688	−0.689	*p* < 0.001
21	−0.751	−0.753	*p* < 0.001
22	−0.362	−0.362	*p* = 0.005
23	−0.695	−0.699	*p* < 0.001

**Table 6 ijerph-20-01734-t006:** Results of the tool reliability assessment.

Item	Alpha if Item Deleted	Item-Total Correlation
2	0.883	0.207
3	0.882	0.236
4	0.875	0.544
5	0.878	0.417
6	0.872	0.613
7	0.882	0.222
9	0.873	0.559
11	0.873	0.558
12	0.886	0.248
13	0.88	0.379
14	0.881	0.28
15	0.877	0.48
16	0.868	0.682
17	0.87	0.65
18	0.874	0.553
19	0.867	0.724
20	0.87	0.65
21	0.868	0.703
22	0.88	0.333
23	0.87	0.65

## Data Availability

Not applicable.

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
