# Peer review of "Polish Adaptation of the *Alarm Fatigue Assessment Questionnaire* as an Element of Improving Patient Safety"

_ijerph, 2023, doi:10.3390/ijerph20031734_

Round 1

Reviewer 1 Report (New Reviewer)

Overall comments

Thank you for the opportunity to review the manuscript for “Polish Cross-Cultural Adaptation of the Alarm Fatigue Assessment Questionnaire as an Element of Improving Patient Safety and Reduction of Occupational Risks”. This is a very important study to ensure questionnaires are validated in the population they are intended for. However, the manuscript in its current form requires a major revision. Please see below for details.

Abstract

The abstract is clear however line 28/29 are not supported by the results and discussion in this manuscript in this format.

Introduction

The introduction provides evidence to support the need for this study. The aim is clearly outlined in this section. 

Materials and Methods

Please provide more detail pertaining to the background of the panel of experts included in the study procedure. This information is pertinent for the weight of the study methodology.

 2.1 Participants and Settings

Please outline how the study recruitment was shared e.g., via email, mail, flyers

Please provide more detail for the inclusion and exclusion criteria. For example, if an individual was a research coordinator with no direct clinical care, did they participate even if they are co located in the ICU space?

How were the study questionnaires collected? – you state they were anonymous - but sent to a member of the research team. Was a box placed in the staff room for people to leave their completed questionnaire in? This needs to be addressed to rule out bias in the study.

How many times was the questionnaire sent out?

2.2 Translation and linguistic validation

Please provide more explanation for the meaning of “high qualifications”, it is not clear what you are intending to say here.

Line 142- it is not clear who performed the forward translation. Please clarify.

Line 147- please clarify how this group were recruited, are they different from the participants in 2.1?

Line 151-153 Pilot study – this requires more explanation? This study was cross sectional in design section 2.1 If this is a new section, a new heading is required, and more detail should be included.

2.3 Cognitive debriefing

Who was involved in this group? How were they recruited? Who ran the group? Please provide a copy of the questions in the Appendix

2.4 Alarm Fatigue Assessment Questionnaire

Line 163 this study is published on the reliability and validity of the questionnaire. The purpose of this manuscript is to produce the Polish version – please revise this section

How were these collected and who entered the data- this needs to be transparent.

2.5 Ethical Consideration

If individuals participated in a focus group session 2.3 then it would not be possible for them to remain anonymous. Please revise this section.  

2.6 Statistical Analysis

Please provide a summary of who in the research team performed this statistical analysis or if this was outsourced. This is required to ensure bias is minimized.

Results

Demographic information included in the table does not need to be included in the text – it is repetitive.

There is no data for anesthesiology which was a target group in section 2.1 – please provided an explanation for their exclusion in the manuscript.

There is no evidence for the results for the cognitive debriefing process.

Line 230 this should be in the discussion section.

Please provide more details as to how the discriminatory power was calculated – this is not transparent

Overall, more detail is required in this section – it is very limited in its current form with all deletions.  

Discussion

The results require revision to further support the argument at the start of the discussion.  There is not enough evidence to support the opening line of the discussion.

As the aim of this study was to produce a Polish adaptation of the Alarm Fatigue Assessment Questionnaire to assess alarm fatigue among healthcare workers in anesthesiology and intensive care units. The rationale for using this questionnaire should not be discussed here. This information belongs in the background. This section requires major revision.

Conclusion:

The conclusion does not reflect the testing of the outcome measure to validate the Polish version. The authors are trying to justify why they conducted the study rather reflecting on the process. This is mostly likely due to a very limited results section. A major review is required.   

Study Limitations

The authors have acknowledged they did not used an electronic version – therefore the methods require a re write to reduce the high probably of bias

Author Response

Dear Reviewer, Thank you for your time and comments. I am enclosing my responses to the issues raised. 
Yours sincerely
Lukasz Rypicz, PhD

Reviewer 2 Report (New Reviewer)

The specificity and sensitivity may need to be calculated for the tool to be fully standardized.

Author Response

Dear Reviewer, Thank you for your time and comments. I am enclosing my responses to the issues raised. 
Yours sincerely
Lukasz Rypicz, PhD

Reviewer 3 Report (New Reviewer)

This study discussed an interesting and important issue, the authors presented the study addresses the fact that the Polish adaptation of a tool investigating alarm fatigue has been published to use in the medical institute so far. However, this Alarm Fatigue Assessment questionnaire created didn't do enough reliability and validity analysis, the sample was underrepresented and most of the research subjects were female.
Although the section on manuscript limitations is partially explained, it is recommended to remind readers that the design of this questionnaire is based on the above research limitations and may only be applicable to nursing staff in medical institutions.
At the same time, it is suggested that the conclusion of this manuscript should be more conservative to avoid excessive use of this Polish version of the Alarm Fatigue Assessment Questionnaire in future research. Also, it is suggested that the title of this article should also be adjusted appropriately.

Author Response

Dear Reviewer, Thank you for your time and comments. I am enclosing my responses to the issues raised. 
Yours sincerely
Lukasz Rypicz, PhD

Round 2

Reviewer 1 Report (New Reviewer)

Thank you for the opportunity to review the revised manuscript for this study. 

Section 2.3 has been removed as per the authors response; however, the numbered headings have not been double checked and are now incorrect. Please review. 

Comment 18: There is no data for anesthesiology which was a target group in section 2.1 – please provided an explanation for their exclusion in the manuscript.Response 18: Anesthesiology has not been excluded. In Polish hospitals we have departments that are called "Anesthesiology and intensive care unit" (it is one department), which is described as ICU. - Please add a comment to the introduction to clarify this for the wider audience. 

The revised version of this manuscript has provided clarity around the methodology and the results. Thank you 

Author Response

Dear Reviewer,
thank you for all your help with which this manuscript has been improved.
The comments from Round II have been fully addressed. 

Lukasz Rypicz

This manuscript is a resubmission of an earlier submission. The following is a list of the peer review reports and author responses from that submission.

Round 1

Reviewer 1 Report

The authors have examined the reliability of a Polish version of the Alarm Fatigue Assessment Questionnaire. This is an interesting and timely study; however, it lacks some critical aspects of cross-cultural adaptation including evaluating the validity, as well as exploratory and confirmatory analyses of the questionnaire.

Author Response

Dear Reviewer,
thank you for your time and insightful review. 
Please find attached your response.
Yours sincerely
Łukasz Rypicz 

Reviewer 2 Report

Thank you for the opportunity to review the manuscript titled "Polish cross-cultural adaptation of the Alarm Fatigue Assessment Questionnaire as an element of improving patient safety and reduction of occupational risks."  I found this manuscript to be very well-written and scientifically sound.  Given the premise that no Polish translation for this instrument exists, this study clearly fills a need for the medical science community.

I have only two minor suggestions for this manuscript:

Results:  Given that the respondents are overwhelmingly female (94%), it may make sense to discuss the gender ratio of the population from which they are drawn.  That is, was the hospital population overwhelmingly female as well?  Why might females have been more likely than males to participate?

Conclusions: The authors begin this section "The presented study shows that no Polish adaption of a tool investigating alarm fatigue has been published so far."  Actually, this study does not show this, but instead responds to the fact that no such adaption exists.  This sentence should be reworded, for example, "The presented study addresses the fact that no Polish adaptation..."

Author Response

(The authors gave the same response as above.)

Reviewer 3 Report

Introduction

This paper investigates the fatigue experienced by personnel in the ICU and anaesthesiology units caused by alarms emitted by medical devices. I can imagine that alarm noises might be upsetting especially when they go off unnecessarily and do not indicate a critical event. Noise pollution is indeed an OHS concern and alarm fatigue could lead to errors and negative consequences for patients. The paper surveys 103 workers in such units in Poland. The authors translated an existing questionnaire into Polish.

Materials and methods

I think there needs to be more information provided about the anonymity of respondents’ data. I was a bit confused about the selection criteria: are the respondents in anaesthesiology and intensive care units only? I notice that item 2 on the questionnaire talks about going to the patient’s bed immediately after I hear an alarm. I’m also not really clear about what the cognitive debriefing session was about. Was this all respondents? the research team? just as a result of the pilot study? What changes were made to the questionnaire, if any, as a result of this debriefing? Were any items added: it looks as though this is not the case.

Results

 Table 2 is very long: could it be an appendix? There doesn’t seem to be a great deal of analysis of the results here. How much of a problem were the alarms to these respondents? This information would add to literature. Items 9 and 10 on the questionnaire seem to be the same. Could you please explain the difference between ventilator alarms and infusion pump alarms? I also was confused about item number 3 concerning limiting the number of alarms. How would this be done? And then there is item 6: “I have a proper professional reaction toward alarms.” What is a proper professional reaction?

Discussion

I’m not sure that the reliability information on the scale should be in this discussion: would this not be better in the results section? At line 193 it is stated… “Implementing specific elements of a safety culture can lead to a reduction in the total number of alarms….”. This seems to be a bit far removed from the study itself. What is a safety culture exactly?

I feel that there needs to be a much broader consideration of why the alarms go off unreliably, which seems to be more of an engineering responsibility rather than an OHS response. I’m also not quite sure that the results of the questionnaire address the core questions. It’s possible that this was not a very good questionnaire to start with. Certainly the results would be stronger if supplemented by interviews about the real basis of the problems caused by the alarms.

This is a fairly modest paper which simply consists of the translation of an existing scale (not very good) and trying it out in Poland.

Author Response

(The authors gave the same response as above.)

Round 2

Reviewer 1 Report

Thank you for your response and for providing the references. Here is a quote from one of the articles you referred to, entitled “Guidelines for the process of cross-cultural adaptation of self-report measures”:

It is highly recommended that, after the translation and adaptation process, the investigators ensure that the new version has demonstrated the measurement properties needed for the intended application. The new instrument should retain both the item-level characteristics such as item-to-scale correlations and internal consistency; and the score-level characteristics of reliability, construct validity, and responsiveness.”

Unfortunately, this manuscript lacks tests of constrcut validity.

Author Response

Dear Reviewer,
I am submitting my response to the Round 2 comments.
Yours sincerely
Lukasz Rypicz

Reviewer 3 Report

Only very minor changes have been made to this manuscript. While the authors have given plenty of responses in their response document, there are very few changes in the article itself. I notice that items 9 and 10 on the questionnaire are still the same but I guess this is a translation issue. I don’t think sufficient changes have been made to make this article acceptable in its present form

Author Response

(The authors gave the same response as above.)
